# Verification of the effect of data-driven brain motion correction on PET imaging

**Hayato Odagiri**[1,2]*, **Hiroshi Watabe**[3], **Kentaro Takanami**[2], **Kazuma Akimoto**[1],
**Akihito Usui**[1], **Hirofumi Kawakami**[4], **Akie Katsuki**[4], **Nozomu Uetake**[4], **Yutaka Dendo**[5],
**Yoshitaka Tanaka**[5], **Hiroyasu Kodama**[5], **Kei Takase**[2], **Tomohiro Kaneta**[1,2]

1 Department of Diagnostic Image Analysis, Tohoku University Graduate School of Medicine, Sendai, Miyagi, Japan, 2 Department of Diagnostic Radiology, Tohoku University Hospital, Sendai, Miyagi, Japan, 3 Division of Radiation Protection and Safety Control, Cyclotron and Radioisotope Center, Tohoku University, Sendai, Miyagi, Japan, 4 GE HealthCare Japan, Tokyo, Japan, 5 Department of Radiology, Tohoku University Hospital, Sendai, Miyagi, Japan

* hayato.odagiri.e2@tohoku.ac.jp

## Abstract

### Introduction

Brain positron emission tomography/computed tomography (PET/CT) scans are useful for identifying the cause of dementia by evaluating glucose metabolism in the brain with F-18-fluorodeoxyglucose or Aβ deposition with F-18-florbetaben. However, since imaging time ranges from 10 to 30 minutes, movements during the examination might result in image artifacts, which interfere with diagnosis. To solve this problem, data-driven brain motion correction (DDBMC) techniques are capable of performing motion corrected reconstruction using highly accurate motion estimates with high temporal resolution. In this study, we investigated the effectiveness of DDBMC techniques on PET/CT images using a Hoffman phantom, involving continuous rotational and tilting motion, each expanded up to approximately 20 degrees.

### Materials and methods

Listmode imaging was performed using a Hoffman phantom that reproduced rotational and tilting motions of the head. Brain motion correction processing was performed on the obtained data. Reconstructed images with and without brain motion correction processing were compared. Visual evaluations by a nuclear medicine specialist and quantitative parameters of images with correction and reference still images were compared.

### Results

Normalized Mean Squared Error (NMSE) results demonstrated the effectiveness of DDBMC in compensating for rotational and tilting motions during PET imaging. In Cases 1 and 2 involving rotational motion, NMSE decreased from 0.15–0.2 to approximately 0.01 with DDBMC, indicating a substantial reduction in differences from the reference image across various brain regions. In the Structural Similarity Index (SSIM), DDBMC improved it to above 0.96 Contrast assessment revealed notable improvements with DDBMC. In

**Data Availability Statement:** All relevant data are within the manuscript files.

**Funding:** HK, AK, and NU received funding in the form of salary from GE Healthcare Japan. HW was funded through a cooperative research agreement with GE HealthCare (grant number 19696262910). The funders had no role in study design, data collection and analysis, decision to publish, or preparation of the manuscript. The specific roles of these authors are articulated in the 'author contributions' section.

**Competing interests:** HK, AK, and NU are employees of GE HealthCare Japan. The study was funded through a cooperative research agreement with GE HealthCare and the software used in the analysis was provided by GE HealthCare. There are no patents, associated with this research to declare. This does not alter our adherence to PLOS ONE policies on sharing data and materials.

continuous rotational motion, % contrast increased from 42.4% to 73.5%, In tilting motion, % contrast increased from 52.3% to 64.5%, eliminating significant differences from the static reference image. These findings underscore the efficacy of DDBMC in enhancing image contrast and minimizing motion induced variations across different motion scenarios.

## Conclusions

DDBMC processing can effectively compensate for continuous rotational and tilting motion of the head during PET, with motion angles of approximately 20 degrees. However, a significant limitation of this study is the exclusive validation of the proposed method using a Hoffman phantom; its applicability to the human brain has not been investigated. Further research involving human subjects is necessary to assess the generalizability and reliability of the presented motion correction technique in real clinical scenarios.

## Introduction

Brain positron emission tomography/computed tomography (PET/CT) is considered useful for understanding pathological conditions that cause cognitive decline, such as changes in neural activity, β-amyloid (Aβ) deposition, and tau aggregation [1–5]. In Japan, F-18-fluoro-deoxyglucose, which evaluate brain glucose metabolism, and F-18-florbetaben and F-18-flute-metamo, which evaluates Aβ deposition, are supplied by pharmaceutical manufacturers. The need for Aβ imaging for evaluating various causes of dementia is increasing. In addition, Tau PET tracers such as F-18-PM-PBB3 and F-18-SNFT-1 are being studied domestically and internationally, with anticipated clinical applications in the future [6,7]. Such preparations will increasingly play an important role in visualizing the accumulation of abnormal proteins such as Aβ and tau in the living brain, detecting their presence or absence, and identifying the cause of dementia [8]. Brain PET scans require varying imaging durations depending on the radio-pharmaceutical used, with 30 minutes for F-18-fluorodeoxyglucose [9,10], 10–20 minutes for F-18-Florbetapir [11], 20–30 minutes for F-18-Flutemetamol [12], and 20 minutes for F-18-Florbetaben [13]. Head motion significantly degrades the quality of reconstructed images, leading to reduced diagnostic value and inaccurate quantification [14]. The most cited effects of patient motion are frame image misalignment, which affects the dynamic analysis of dynamic protocols, and resolution loss due to motion blurring in the frame. Moreover, motion during scans can cause mis-estimation in tracer kinetic modeling, introducing inaccuracies in the quantitative analysis of the data [15]. Several techniques have been implemented to compensate for head motion, including hardware-based motion tracking methods that utilize an external device to track a marker attached to the head [15,16], and marker-less tracking methods that employ a charged-coupled device camera to extract the surface shape of the object [17,18]. The motion tracking method using Polaris Vicra requires the attachment of light-reflective markers to the patient, which is time-consuming to set up and may cause the attached markers to come off due to movement [19].

A data-driven brain motion correction (DDBMC) technique has been developed [20–22]. In DDBMC, image reconstruction is performed with ultra-short frames (≤1 second) from list-mode data. Motion is estimated by performing image rigid registration between each frame and a selected reference frame, allowing for motion-corrected listmode reconstruction. Each frame is reconstructed using maximum-likelihood expectation maximization without subsets,

because of the low number of events in each frame. Since accurate quantification is not crucial, scatter correction and point spread function modeling are omitted, and only a few iterations are carried out. If necessary, you can use the standard attenuation correction map generated from CT images [5,23]. In the present study, we validated the method using PET/CT and the Hoffman brain phantom, assuming sustained rotational and tilting motion with a range of approximately 20 degrees. Although there have been reports in real patients, these reports have been limited to motion angles of about 5 degrees and have not been validated for large or sustained movements, thus validating the accuracy of the more complex correction effects.

## Materials and methods

### PET/CT system

In this study, the Discovery MI PET/CT system (GE HealthCare, Milwaukee, WI, USA) was used. It consists of a 128-slice CT system and a 4-ring PET system with lutetium-yttrium oxy-orthosilicate (LYSO) crystals and silicon photomultiplier-based (SiPM) detectors providing a 20-cm axial field of view and a 70-cm transaxial field of view. The system consists of a total of 19,584 crystals and 9792 SiPM channels. Each crystal has dimensions of 3.95 x 5.3 x 25 mm$^3$ and is connected to a light guide to optimize light collection, thereby improving sensitivity and resolution. The LYSO and SiPM detectors enable time-of-flight compatibility with time resolution below 380 ps [24,25].

### Hoffman phantom imaging

The Hoffman 3D brain phantom (AcroBio Corporation, Tokyo, Japan) was used in this study. Fig 1 shows the Hoffman 3D Brain phantom installation. The Hoffman phantom was placed on a rotating and tilting table. Rotational motion of the head about the scanner axis and tilting motion of the head (i.e., rotations about the patient's left-right axis) were used to simulate head movements during PET imaging. For continuous rotation, a syringe driver was used to rotate the phantom at a constant velocity (2.5 degrees/minute). For ground truth tracking of motion, the Polaris Vicra optical tracking device (Northern Digital Inc, Ontario, Canada) was used. Three markers were attached to the parietal side of the Hoffman phantom. Two charge-coupled device cameras and infrared light were used to measure movement during PET imaging.

The acquisition and movement protocol for phantom imaging is shown in Fig 2. First, imaging was performed to create a reference image. The $^{18}$F solution in the Hoffman phantom was adjusted to 37 MBq at the start of imaging [26]. With the phantom fixed, CT for

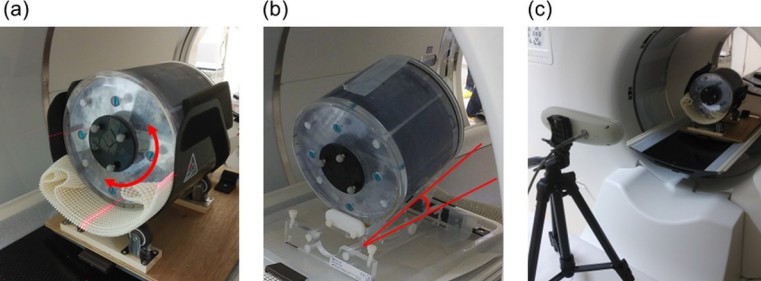

**Fig 1. Deployment of the Hoffman 3D brain phantom and an optical tracking device.** (a) Simulated rotational motion of the head. (b)Simulated tilting motion of the head. (c)Infrared reflective marker and Polaris Vicra optical tracking device.

1. Reference scan (no motion)

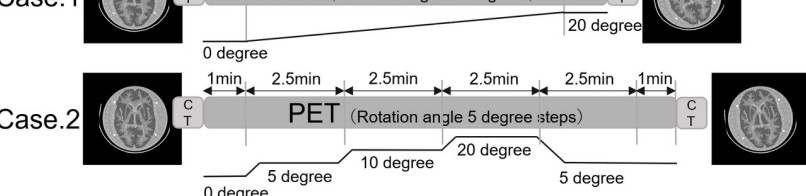

2. Rotational motion scan

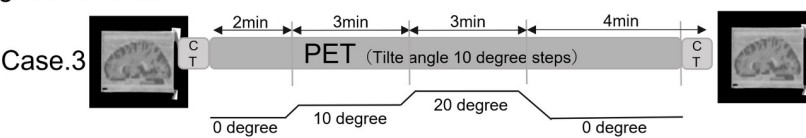

**Fig 2. Outline of movements during PET scans.**

attenuation correction was performed, followed by PET imaging in listmode over 8 minutes. Next, imaging with a rotating motion was performed. In the first case, PET imaging was performed with the phantom being stationary for 1 minute after attenuation -corrected CT imaging. Next, the phantom was rotated slowly to the right and continuously rotated up to 20 degrees over 8 minutes (Case 1).

PET/CT imaging was performed in a stationary state, which was defined as the reference. In Case 1, the starting point of imaging was 0 degrees, from which the patient was slowly and continuously rotated to the right, up to 20 degrees. In Case 2, the starting point of imaging was 0 degrees. Rightward rotation of 5 degrees, 10 degrees, and 20 degrees was applied. Finally, it returned to the original position by rotating it to the left from the position of 20 degrees to the right. In Case 3, tilting motion of 0, 10, and 20 degrees was applied in a stepwise manner before returning to 0 degrees. Since different bases were used for rotational and tilting motions, a reference scan was performed just before imaging in Case 3 for comparison.

In the second case, after attenuation-corrected CT imaging, the phantom was rotated 5, 10, or 20 degrees to the right in 1-minute increments of stationary PET imaging and then rotated 15 degrees to the left at approximately 9 minutes after the start of imaging. (Case 2).

Next, the tilting motion was created in a stepwise manner from 0 to 10 degrees and then from 10 degrees to 20 degrees. The tilt was then gradually lowered back to 0 degrees (Case 3). Since the tilting motion used a base to reproduce vertical movement that was different from the one used for rotational motion, one reference scan was performed just before the imaging for Case 3. Images from Case 3 were compared with the reference image from that scan.

## Data-driven brain motion correction

We processed motion correction using the lmDuetto PET toolbox Ver.02.18 (GE HealthCare). DDBMC makes ultra-short (approximately 1 second/frame, depending on the data) reconstructions from listmode data and estimates motion between these ultra-short images and a

reference image at the beginning of acquisition (30 seconds/frame) using rigid registration [20]. For reconstruction of the ultra-short frames, the maximum likelihood expectation maximization method was used. Reconstructions were performed with six iterations, with no attenuation or scatter corrections [5]. After estimation, a fully motion corrected listmode reconstruction can be performed [21]. Reconstruction with motion correction was performed using Q.Clear™, a Bayesian penalized likelihood reconstruction algorithm that incorporates a penalty factor (a relative difference prior) to suppress noise. β, the only user input, controls the relative strength of the noise regularizing term. Q.Clear™ incorporates time-of-flight and point spread functions for resolution recovery. Reconstruction was performed with a field of view of 300 mm, a matrix size of 256, pixel size of 2.73 mm, and a β value of 200 [27–29].

## Image analysis

We used Prominence Processor (Nihon Medi-Physics Co., Ltd., Tokyo, Japan) for image analysis. The normalized mean squared error (NMSE) shows the error between the reference image and the image with and without motion correction, with a value closer to zero indicating a smaller difference from the reference image. The reference image was taken before rotational and tilting motion, respectively. Here, x and y are the number of counts in the matrix, respectively, z is the number of slices, f(i, j) is the reference image, and g(i, j) is the number of pixels in the target image. Of the total of Seventy-one slices from one-bed scan imaging, it evaluated 43 slices delineated from the parietal lobe to the cerebellum.

$$NMSE = \frac{\sum_{i=0}^{x} \sum_{j=0}^{y} \left(g(i,\,j) - f(i,\,j)\right)^2}{\sum_{i=0}^{x} \sum_{j=0}^{y} f(i,\,j)^2}$$

Furthermore, a structure similarity index (SSIM) was used to measure the degree of similarity between the reference image and an image of interest [30]. The SSIM is sensitive to contrast, luminance, and structures within an image, with values ranging from 0 to 1. As SSIM approaches 1, the similarity with the reference image increases. Of the total of Seventy-one slices from one-bed scan imaging, it evaluated 43 slices delineated from the parietal lobe to the cerebellum. SSIM calculations were performed using skimage 0.21.0 and Python 3.8.10.

A total of 41 regions of interest were defined in the gray and white matter across 5 slices from the parietal lobe to the cerebellum on the PET images (Fig 3). The mean counts within these regions of interest (ROI) were obtained for both gray matter ($GM_P$) and white matter ($WM_P$). Given that the gray matter to white matter ratio ($GM_d/WM_d$) is defined as 4 on the digital phantom image for gray matter ($GM_d$) and white matter ($WM_d$), the % contrast is calculated as follows:

$$\% \text{ contrast} = \frac{GM_P/WM_P - 1}{GM_d/WM_d - 1} \times 100$$

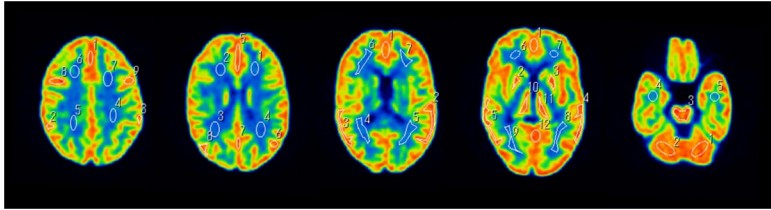

**Fig 3. Evaluation of % contrast.**

A total of 41 regions of interest were defined in the gray and white matter of 5 slices from the parietal lobe to the cerebellum for images reconstructed with Q.Clear™; the beta parameter was 200.

For blinded visual evaluation, the images were rated on a five-point scale as to whether they could be used for diagnosis, assuming that the patient had dementia. Readers were nuclear medicine specialists, consisting of two doctors and three radiologists. Visual evaluations were performed on three cross-sectional image displays using Xeleris (GE HealthCare). The Dunnett's test and Wilcoxon signed-rank test statistical analyses were used with JMP Pro, version 17.1.0 (SAS Institute Inc., Cary, NC, USA).

# Results

## Visual evaluation

Fig 4 shows the motion plots used in the ultra-short (1 second) DDBMC for Polaris Vicra optical tracking and listmode data in each case. The top row shows motion plots obtained from Polaris Vicra external devices. The bottom row is a motion plot of displacements computed from ultrashort frame images of the listmode data. Fig 4(A) shows the relationship between displacement in the X, Y, and Z directions and elapsed time, and Fig 4(B) shows the relationship between angular displacement in the X, Y, and Z directions and elapsed time. Comparison of motion corrected images from data of both methods resulted in the mean values (standard deviation) of NMSE: 0.005 (0.001) for Case 1, 0.008 (0.002) for Case 2, and 0.011 (0.003) for Case 3. We confirmed that displacement calculations comparable to Polaris Vicra optical tracking are possible even from ultrashort (1 second) images of listmode data.

Reconstructed images with and without DDBMC for rotational motion and tilting motion are shown in Figs 5 and 6, respectively. In images with rotation and no DDBMC, there were

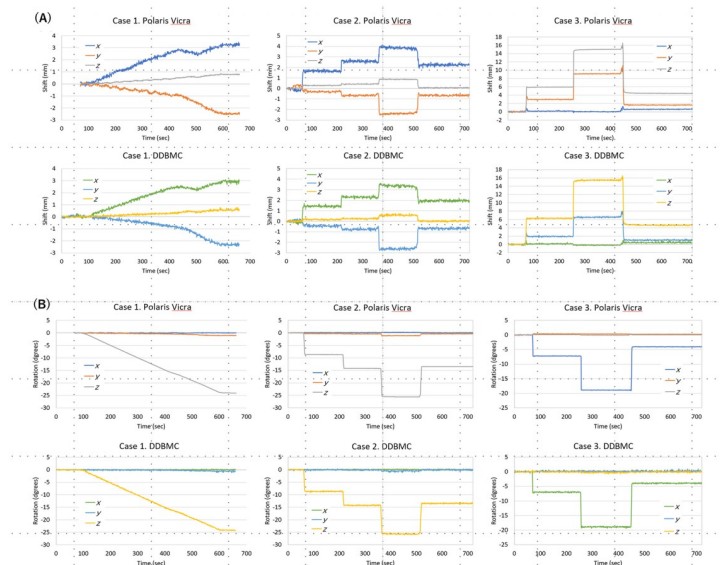

**Fig 4. Motion plots obtained from the Polaris Vicra optical tracking and DDBMC.** Fig 4 shows the motion plots used in the ultra-short (1 second) DDBMC for Polaris Vicra optical tracking and listmode data in each case. (A) shows the relationship between displacement in the X, Y, and Z directions and elapsed time, where the vertical axis is displacement (mm) and the horizontal axis is time (seconds). (B) shows the relationship between elapsed time and angular displacement in the X, Y, and Z directions, with angle (degrees) on the vertical axis and time (seconds) on the horizontal axis.

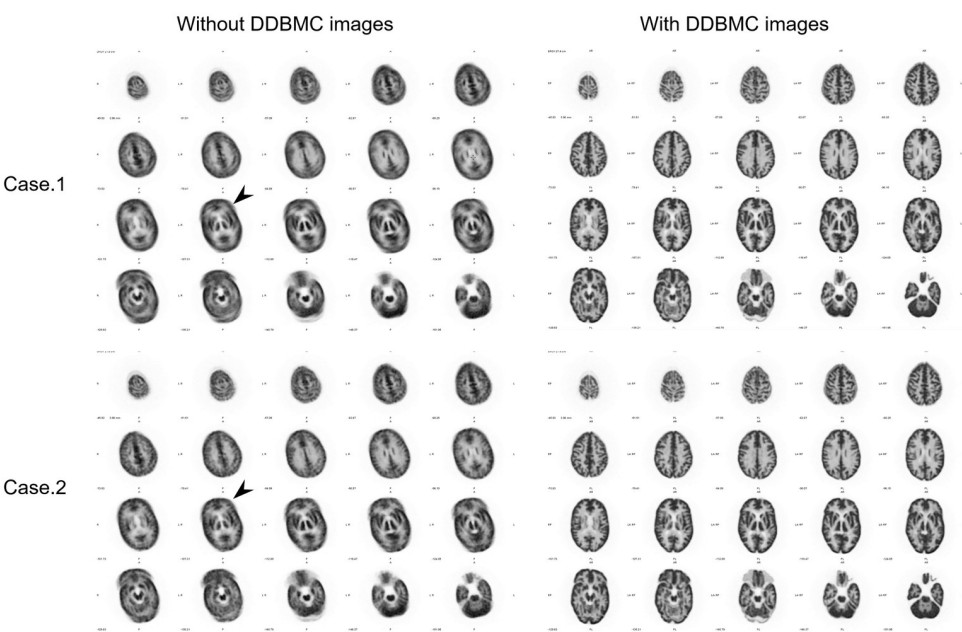

**Fig 5. Images with and without DDBMC for data containing rotational motion.** The left images are without DDBMC and the right images are with DDBMC. Without DDBMC, the brain was tilted to the right and the entire brain was blurry (►). With DDBMC, the brain structure was clearer and the contrast between gray and white matter appeared improved.

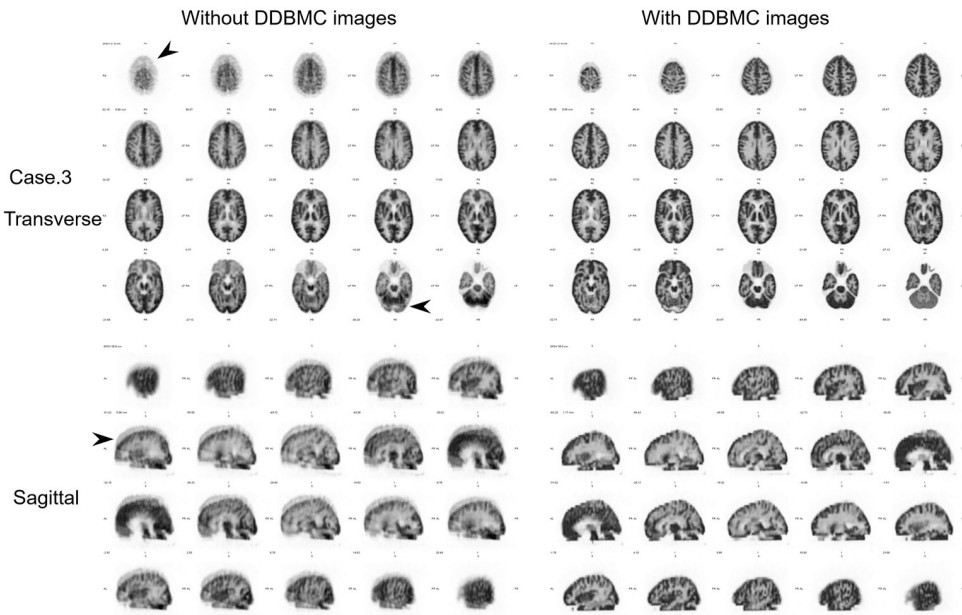

**Fig 6. Images with and without DDBMC for data containing tilting motion.** In the left images without DDBMC, the structures of the basal ganglia were clear in the transverse sections and the contrast between gray and white matter was preserved, making it difficult to discern the effects of movement. In the sagittal sections, the influence of motion could be observed in the frontal lobe (►). In the right images with DDBMC, the displacement observed in the sagittal slices disappeared and the effect of correction could be seen.

**Table 1. Visual assessment scores by professional doctors and radiologists.**

| | | Evaluators | | | | | Average score | p-value |
|---|---|---|---|---|---|---|---|---|
| | | **A** | **B** | **C** | **D** | **E** | | |
| Without DDBMC | Ref image | 4 | 4 | 4 | 4 | 4 | 4 | — |
| | Case1 | 1 | 0 | 0 | 0 | 0 | 0.2 | p<0.05※ |
| | Case2 | 1 | 1 | 0 | 2 | 1 | 1 | p<0.05※ |
| | Case3 | 3 | 3 | 1 | 3 | 3 | 2.6 | p<0.05※ |
| With DDBMC | Case1 | 4 | 4 | 4 | 4 | 4 | 4 | 1 |
| | Case2 | 4 | 4 | 4 | 4 | 4 | 4 | 1 |
| | Case3 | 4 | 4 | 4 | 4 | 4 | 4 | 1 |

※Wilcoxon signed-rank test (p<0.05).

DDBMC: Data-driven brain motion correction.

no visual differences between Case 1, which included continuous motion, and Case 2, which included gradual motion. In both cases, the brain was tilted to the right, giving the impression that the entire brain was blurry. In images with DDBMC, brain structures were clearer, the boundary between the gray and white matter was more distinct, and image quality was higher. In Case 3 images with vertical motion and no DDBMC, structures were clear in the transverse slice at the level of the basal ganglia and ins cross-sectional images; they did not seem to contain motion blurring. However, displacement could be seen in the transverse image of the parietal lobe and cerebellum. In sagittal images, movement in the frontal region could be seen. The effect of movement could be recognized in multiple cross-sections. In images with DDBMC, the displacement that had occurred between the parietal and cerebellar sides was eliminated; the structure of the entire brain was clearer.

The results of the visual evaluation are shown in Table 1. Two nuclear medicine physicians and three certified nuclear medicine technologists rated the images on a five-point scale to determine whether the PET scan images were adequate for diagnosing dementia. The five raters evaluated the images on a scale of 0 to 4, corresponding to "not diagnostic", "somewhat non-diagnostic", "neither diagnostic nor non-diagnostic", "somewhat diagnostic", and "diagnostic", respectively. In images without DDBMC, the average scores were 0.2 for Case 1 and 1.0 for Case 2, indicating "Not diagnostic". For Case 3, the average score was 2.6. All evaluators judged the reference images and images with DDBMC as "Diagnostic". A comparison of reference images and other images showed significantly lower values for images without DDBMC, while no significant differences were found between reference images and images with correction. The effect of DDBMC was also confirmed with the visual evaluation.

## Normalized mean squared error and structure similarity index

The results of NMSE for each slice, comparing images with rotational or tilting motion to the reference images taken in a static state, are presented in Fig 7. Additionally, Table 2. provides the mean and standard deviation values of NMSE and SSIM for all slices. In the case of images with rotational motion (Cases 1 and 2), NMSE ranged from 0.15 to 0.2 from the basal ganglia to the cerebellum, indicating a significant difference from the reference images. With DDBMC, the variation based on slice position and reduces differences from the reference images. With DDBMC, mean NMSE decreases to approximately 0.01 for both Cases 1 and 2. In terms of SSIM, there is a significant improvement, with Case 1 increasing from 0.892 to 0.968, and Case 2 increasing from 0.860 to 0.967. In Case 3, involving tilting motion, the

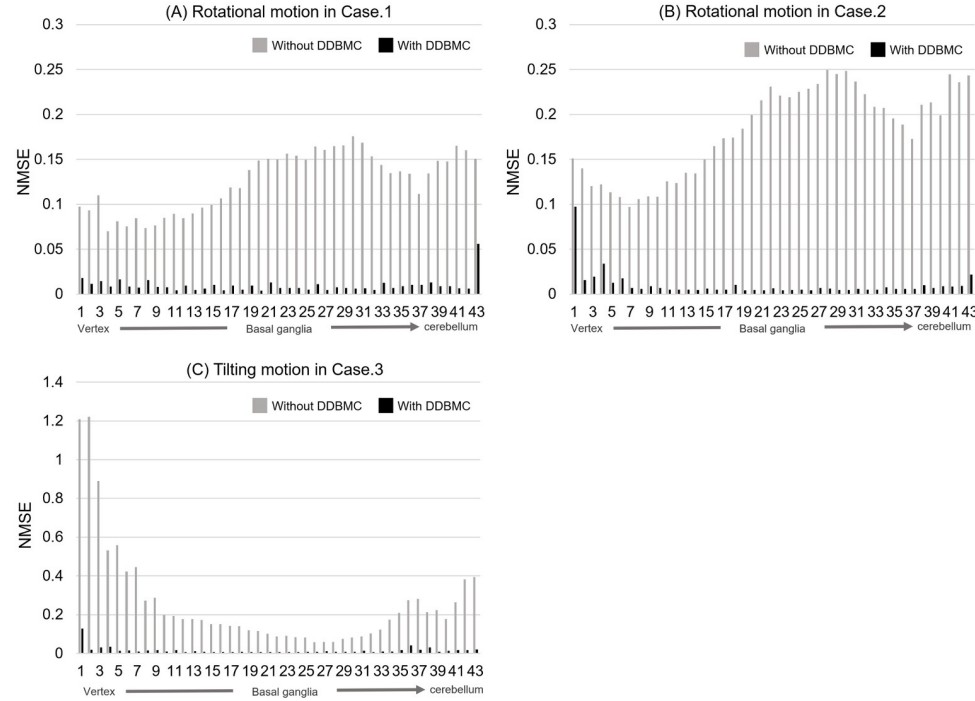

**Fig 7. Normalized mean square error in transverse images.** The normalized mean square error (NMSE) results for reference images in the static state and images with each motion, compared with and without DDBMC, are shown. (A) depicts Case 1 with continuous rotational motion, (B) shows Case 2 with stepwise rotational motion, and (C) illustrates Case 3 with tilting motion. The closer the NMSE value on the vertical axis to 0, the closer the image is to the reference image.

differences in NMSE values in the basal ganglia were smaller than those in rotational motion. However, the NMSE values in the vertex and cerebellar regions were larger. With DDBMC, NMSE significantly improved from 0.262 to 0.016, and SSIM increased from 0.874 to 0.962.

## Contrast assessment

The results of the % contrast are presented in Table 3. The reference image taken in the static state had 74.3% contrast. In Case 1, which includes continuous rotational motions, % contrast

**Table 2. Normalized mean square error and structural similarity index with and without DDBMC.**

|  |  | Without DDBMC | With DDBMC | p-value |
|---|---|---|---|---|
| NMSE | Case1 | 0.126 (0.033) | 0.010 (0.008) | p<0.05※ |
|  | Case2 | 0.182 (0.049) | 0.010 (0.018) | p<0.05※ |
|  | Case3 | 0.262 (0.269) | 0.016 (0.019) | p<0.05※ |
| SSIM | Case1 | 0.892 (0.004) | 0.968 (0.004) | p<0.05※ |
|  | Case2 | 0.860 (0.005) | 0.967 (0.005) | p<0.05※ |
|  | Case3 | 0.874 (0.003) | 0.962 (0.003) | p<0.05※ |

mean values (standard deviation)

※student t-test (p<0.05).

NMSE: Normalized mean square error. SSIM: Structure similarity index. DDBMC: Data-driven brain motion correction.

**Table 3. Comparison of % contrast between images with and without DDBMC and reference images.**

|  |  | % contrast | p-value |
|---|---|---|---|
|  | Ref- image | 74.3 (9.6) | — |
| Without DDBMC | Case1 | 42.4 (4.9) | p<0.05※ |
|  | Case2 | 45.8 (5.9) | p<0.05※ |
|  | Ref- image 2 | 68.8 (9.5) | — |
|  | Case3 | 52.3 (11.3) | p<0.05※ |
| With DDBMC | Ref- image | 74.3 (9.6) | — |
|  | Case1 | 73.5 (11.4) | n.s |
|  | Case2 | 69.6 (11.3) | n.s |
|  | Ref- image 2 | 68.8 (9.5) | — |
|  | Case3 | 64.5 (8.3) | n.s |

mean values (standard deviation)

※Dunnett test (p<0.05).

DDBMC: Data-driven brain motion correction.

was 42.4%, which improved to 73.5% with DDBMC. In Case 2, which included a stepwise rotational motions, % contrast improved from 45.8% to 69.6% with DDBMC. In Case 3, which includes tilting motion, % contrast of the reference image was 68.8%, while it was 52.3% without DDBMC and 64.5% with DDBMC. With DDBMC, % contrast improved in all cases and was no longer significantly different from % contrast of the reference image.

## Discussion

Since head movement during brain PET imaging can cause image degradation, the head is fixed during imaging. Because head movements during brain PET imaging can lead to image degradation, the head is immobilized during imaging. However, due to the extended time required for image acquisition, it can be challenging for all subjects to remain completely still, depending on their age and physical condition. In short-duration imaging, there is a potential for decreased resolution. Therefore, techniques are needed to minimize the effect of head motion on the images. Keller et al. employed an optical motion tracking system that captures head movements from the outside by attaching markers to the patient's head [16]. Slipsager JM et al. demonstrated the utility of a motion tracking system based on computer vision technology, employing a structured optical surface scanner. The system utilizes a synchronized optical modulator and camera for continuous scanning of the patient's face. In contrast to marker-based solutions, this approach eliminates the need for attaching optical markers, resulting in a reduction in clinical preparation time [31]. The method of attaching an external device directly to the subject's head or a tracking method involving surveillance cameras might require more preparation time and could be more burdensome for the subject. In addition, if the marker attached to the subject's head becomes dislodged or out of the tracking field of view, tracking becomes difficult and accurate motion estimation becomes impossible. In this study, we used DDBMC, which was devised as a correction method that does not require an external device, to verify the effect of data-driven motion correction with rotational and tilting motions of the head during brain PET imaging.

DDBMC uses image-based rigid registration to the reference image from ultra-short time frame reconstructed images (approximately 1 second/frame) with data collected in listmode [20,21]. The estimated motion was compared to those tracked by the Polaris external optical

tracking device, which is considered the gold standard for tracking movement [5]. Even though the frames were short (approximately 1 second/frame) and were thus very noisy, the motion calculated with DDBMC matched that of the external optical device with high accuracy. Rotational speed was different for sustained and stepwise rotational motions, but we believe that the rotational speed used in the present study can be sufficiently corrected. In addition, in the count volume obtained from the radioactivity concentrations used in this study, it was possible to calculate the amount of movement in a very short time of approximately 1 second/frame. Since reflexive movements are expected in actual clinical practice, verification of how much movement speed can be handled without error is an issue for future investigation.

In the NMSE evaluation of rotational motions, misalignment in the basal ganglia was found to be as large as 0.1 to 0.20 without DDBMC for continuous and phasic movements. Misalignment in the parietal and cerebellar area were as large as 0.4 to 1.2 for tilting motions. With tilting motions, there were slices with NMSE less than 0.1 at the level of the basal ganglia despite the presence of movement. This indicates that it is difficult to recognize misalignments because they cannot be captured as large changes in structures in cross sections at the level of the basal ganglia, compared to the parietal or cerebellar side. In the visual evaluation, the uncorrected image had a higher value (2.6) for Case 3 than the other uncorrected images. This indicates that there are situations in which the amount of movement is small or it is difficult to visually determine whether the image contains movement due to the way it moves. Therefore, we recognize the need for visual confirmation in multiple cross-sections. Furthermore, the SSIM with DDBMC showed a significantly higher value of 0.96 or higher than that without DDBMC. We believe that this result is consistent with the fact that all cases after DDBMC as "Diagnostic" in the visual evaluation by the specialist physicians. Spangler-Bickell et al. reported that motion compensation has no effect on images even in the absence of motion [5,21]. It would be desirable to incorporate motion correction into all brain PET scans.

In the % contrast analysis, DDBMC led to improvements to the point where corrected images were not significantly different from reference images, although slightly lower than the reference image. This may be due to the difference between the start time of the reference image and the start time of images with motion, which might have been caused by the count difference due to radioactive decay. By capturing a reference image and an image containing motion in a sequence, we assume that the % contrast will be closer to that of the reference image after DDBMC.

Tracking correction methods using external devices reported in previous studies have some drawbacks, such as complicated marker attachment, burden on the subject, and markers falling off or moving out of the field of view during tracking [32]. DDBMC in this study was completely device-less and thus utilized listmode data obtained during a routine examination without imposing additional burden on the subject. When verifying of the effect of DDBMC on rotational motion and vertical motion using the Hoffman 3D brain phantom, the correction effect was confirmed relative to the reference image based on NMSE and % contrast. Although it is difficult to determine the detection limit of displacement from DDBMC based on this study alone, we believe that it is sufficient to compensate for displacements of approximately 2.5 degrees/minute and 4.5 mm/minute, based on the maximum 20 degrees of continuous rotation over 8 minutes with a Hoffman 3D brain phantom of 206 mm in diameter.

There are other limitations to this study. We evaluated the range of motion of 20 degrees of rotation and 20 degrees of tilting motions. In clinical examinations, head motion involves a combination of six degrees of freedom (6-DOF). Spangler-Bickell et al. reported the effectiveness of DDBMC even in clinical cases [21]. This suggests that DDBMC is useful for various

movements. However, understanding the limits of the correction region for movements based on the combination of 6-DOF is crucial, and further verification is deemed necessary in the future. We believe that more complex movements should be considered. As a limitation of this study, since DDBMC requires a still image of about 30 seconds as a reference image, it may not be possible to ensure accuracy with an image that is constantly moving.

Furthermore, image quality proved to be sufficient for the diagnosis of dementia based on visual evaluation by specialists. However, there can be complex movements that are not anticipated in actual clinical practice. Since the correction effect might vary depending on the frequency and amount of movement, we believe it is important to understand the limits of the correction effect in advance. The results of this study mainly apply to static data, where changes in tracer distribution over time are minimal. In dynamic data, which captures changes in tracer distribution, it is necessary to update the reference image regularly. The optimal strategy for handling this is required. Revilla et al. proposed a data-driven method, the three-dimensional center of distribution, reporting significant reduction in blurring due to head motion [23]. It has been noted that this method may not be able to accommodate large changes in tracer activity within the brain. Verification is needed in the present study regarding the DDBMC examined here in cases of significant changes in tracer activity within the brain. Future studies will examine the effect of motion compensation on more movements encountered in clinical practice and will assess the impact on standardized uptake values in segmented regions of the brain. Additionally, Zeng T et al. introduced a novel structure for deep neural networks for head motion estimation using supervised learning [33,34]. Tumpa TR et al. also proposed an unsupervised deep learning approach for brain motion correction through 3D image registration [35]. While keeping an eye on these new technologies, we aim to promote the utilization of DDBMC in clinical examinations.

## Conclusions

The head motion by the Hoffman phantom was validated by moving the head up to approximately 20 degrees in rotational angle and tilt. DDBMC technique is a promising image reconstruction tool that can effectively compensate for large angular rotational and tilting motion of the head, eliminating the need for rescanning and degrading image quality. This has potential clinical applications in brain PET imaging for the diagnosis of dementia. Further studies with human subjects are needed to assess the generality and reliability of the presented motion compensation technique in real clinical situations.

## Acknowledgments

The authors thank the nuclear medicine laboratory staff of the Division of Radiology, Department of Medical Technology, Tohoku University Hospital for their valuable support.

## Author Contributions

**Conceptualization:** Hayato Odagiri, Hiroshi Watabe, Kentaro Takanami.

**Formal analysis:** Hayato Odagiri, Hiroshi Watabe, Kentaro Takanami, Kazuma Akimoto, Akihito Usui, Yutaka Dendo, Yoshitaka Tanaka.

**Funding acquisition:** Kentaro Takanami.

**Investigation:** Hiroyasu Kodama.

**Methodology:** Hayato Odagiri, Hiroshi Watabe, Yutaka Dendo.

**Project administration:** Hayato Odagiri, Hirofumi Kawakami, Akie Katsuki, Nozomu Uetake, Kei Takase, Tomohiro Kaneta.

**Software:** Hiroshi Watabe, Hirofumi Kawakami, Akie Katsuki, Nozomu Uetake.

**Supervision:** Hiroshi Watabe, Tomohiro Kaneta.

**Validation:** Hiroshi Watabe, Kentaro Takanami, Akihito Usui, Yoshitaka Tanaka, Hiroyasu Kodama, Tomohiro Kaneta.

**Visualization:** Kentaro Takanami, Kazuma Akimoto, Tomohiro Kaneta.

**Writing – original draft:** Hayato Odagiri.

**Writing – review & editing:** Hayato Odagiri, Hiroshi Watabe, Kentaro Takanami, Kazuma Akimoto, Akihito Usui, Yutaka Dendo, Tomohiro Kaneta.

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
