## [Decision Letter · Decision Letter 0]

30 Jan 2024

PONE-D-24-01135Verification of the effect of Data-Driven Brain Motion Correction on PET imagingPLOS ONE

Dear Dr. Odagiri,

Thank you for submitting your manuscript to PLOS ONE. After careful consideration, we feel that it has merit but does not fully meet PLOS ONE’s publication criteria as it currently stands. Therefore, we invite you to submit a revised version of the manuscript that addresses the points raised during the review process.

We look forward to receiving your revised manuscript.

Kind regards,

Khan Bahadar Khan, Ph.D

Academic Editor

PLOS ONE

Journal Requirements:

"The study was funded through a cooperative research agreement with GE HealthCare and the software used in the analysis was provided by GE HealthCare.（Grant number 19696262910，Principal Investigator：HW）

HK, AK, and NU are employees of GE HealthCare Japan.

HK, AK, and NU are employees of GE HealthCare Japan."              

"Authors with competing interests

Enter competing interest details beginning with this statement:

I have read the journal's policy and the authors of this manuscript have the following competing interests:

HK, AK, and NU are employees of GE HealthCare Japan.

The study was funded through a cooperative research agreement with GE HealthCare and the software used in the analysis was provided by GE HealthCare."

4. In this instance it seems there may be acceptable restrictions in place that prevent the public sharing of your minimal data. However, in line with our goal of ensuring long-term data availability to all interested researchers, PLOS’ Data Policy states that authors cannot be the sole named individuals responsible for ensuring data access (http://journals.plos.org/plosone/s/data-availability#loc-acceptable-data-sharing-methods).

5. Please upload a new copy of Figures 4a, 4b   as the detail is not clear. Please follow the link for more information: " ext-link-type="uri" xlink:type="simple">https://blogs.plos.org/plos/2019/06/looking-good-tips-for-creating-your-plos-figures-graphics/"
https://blogs.plos.org/plos/2019/06/looking-good-tips-for-creating-your-plos-figures-graphics

Reviewers' comments:

Reviewer's Responses to Questions

**Comments to the Author**

1. Is the manuscript technically sound, and do the data support the conclusions?

Reviewer #1: Partly

Reviewer #2: Yes

2. Has the statistical analysis been performed appropriately and rigorously? 

Reviewer #1: I Don't Know

Reviewer #2: Yes

3. Have the authors made all data underlying the findings in their manuscript fully available?

Reviewer #1: Yes

Reviewer #2: Yes

4. Is the manuscript presented in an intelligible fashion and written in standard English?

Reviewer #1: No

Reviewer #2: Yes

5. Review Comments to the Author

Reviewer #1: The spatial resolution of the PET scanner is constantly improving. Head motion could degrade the image quality and get more and more notice. Camera system such Polaris Vicra was proven to have good measurement performance but requires the attachment of a marker to the subject’s head. Data-driven motion correction is promising as it does not require additional hardware but need to improve the accuracy. This paper validated an existing correction method on phantom studies with simulated movement patterns. Reconstructed images with and without correction were compared both quantitatively and qualitatively. Compared to the reference image, contrast and error were improved. Despite these investigations, I do have comments for you to consider.

1. Abstract. A big limitation of the study is that the method was only validated with Hoffman phantom but not human brain. This should be stated explicitly.

2. Abstract. “% contrast and normalized mean squared error were improved after correction.” Numbers would talk more than simply stating “improved”.

3. Abstract. “Although the effectiveness of brain motion correction was confirmed in this experiment, it is necessary to understand the relationship between the range of motion and limitations in brain motion correction processing because there might be more complex movements in clinical practice.” I consider you are not saying results. Please move to conclusions.

4. Introduction. “tau PET imaging preparations are also being investigated.” What is “imaging preparations”? Please define.

5. Introduction. “Movement during the examination might result in inaccurate images.” An image cannot be inaccurate, only its quantification can.

6. Introduction. “Research and development of technologies to compensate for head motion is underway, including hardware-based motion tracking methods that track markers attached to the head with an external device…” Hardware based motion measure and compensation methods have been mature for more than a decade. So it is not appropriate to say they are still underway.

7. Introduction. “charged-coupled device（CCD）” No need to state the abbreviation if it is only mentioned once.

8. Introduction. “A data-driven brain motion correction (DDBMC) technique…” Although reader can find implementation details themselves, it is still helpful to introduce how it works briefly.

9. Introduction. “In the present study, we validated our method…” Consider change “our” to “the” as it is not proposed by the authors.

10. Materials and methods. GE HealthCare - GE Healthcare

11. Lutetium based scintillators (LBSs). Do you mean LSO or something else?

12. “The LBSs and SiPMs detectors enable time resolution below 380 ps and time-of-flight compatibility.” Consider rephrasing it to “The LBSs and SiPMs detectors enable time-of-flight compatibility with time resolution below 380 ps.” I did not go through the whole paper, but it is likely other sentences need to be rephrased too.

13. “Although the range of motion is considered sufficient, it is assumed that a variety of movements will be mixed in actual.” Although you simulated three cases of movement, they still cannot represent the real case. For example, a real head movement can be the combination of six DOFs. Can you comment on this?

14. Line 133, 1 second/frame was used. Have you performed denoising? “list data” should be “listmode data”.

15. Contrast and NMSE are good metrics to evaluate the correction. It would be nice to see SSIM to reflect the recovery of global features.

16. “The results of the visual evaluation are shown in Table 1” Four average scores are 4, which suggests all raters have consensus of scoring 4? Without is out of boundary.

17. “The closer NMSE is to 0, the closer the image with motion is to the reference image” Consider remove this sentence, as it already appeared in Methods.

18. Discussion. “However, it is difficult for the subject to remain still because of the long time needed to acquire images”. New scanners can scan head within a minute. The associated challenge will then be the vulnerability of ultra-high resolution.

19. “In this study, we used DDBMC, which was devised as a correction method that does not require an external device.” Some important references are missing. There are other data-driven method papers in recent years. For example:

Rezaei et al. PMB 2021, Rigid motion tracking using moments of inertia in TOF-PET brain studies.

Sun T. et al. PMB 2022, An iterative image-based inter-frame motion compensation method for dynamic brain PET imaging.

20. A limitation is the study was only validated in static scan but not dynamic scan.

Reviewer #2: This paper focuses on addressing the challenge of motion artifacts in brain PET scans, which can interfere with an accurate diagnosis of dementia. The study applied a data-driven brain motion correction method and demonstrated its effectiveness in compensating for head movements during imaging. By utilizing a Hoffman phantom to replicate realistic motions, the paper presents compelling results showing improved % contrast and reduced normalized mean squared error after correction for head motion. Visual evaluations by a nuclear medicine specialist also support the efficacy of the motion correction technique, suggesting its potential for enhancing the diagnostic quality of PET imaging in clinical settings. However, the study acknowledges the need for further exploration regarding the range of motion and potential limitations in more complex clinical scenarios. In addition, the introduction is not clear for other motion correction methods. Overall, the paper contributes valuable insights and practical implications for refining brain imaging techniques and advancing the field of PET/CT in neuroimaging.

Comment 1:Page 4, line 78. Reference 15 is not related to hardware-based motion tracking. I suggest changing appropriate reference here.

[1] Zeng, Tianyi, et al. "Markerless head motion tracking and event-by-event correction in brain PET." Physics in Medicine Biology 68.24 (2023): 245019.

[2] Jin, Xiao, et al. "List-mode reconstruction for the Biograph mCT with physics modeling and event-by-event motion correction." Physics in Medicine Biology 58.16 (2013): 5567.

[3] Iwao, Yuma, et al. "Brain PET motion correction using 3D face-shape model: the first clinical study." Annals of Nuclear Medicine 36.10 (2022): 904-912.

Comment 2: In the Introduction section, the author did not introduce the data-driven method, and there is no review for other data-driven head motion correction methods. Such as centroid of distribution.

Reference: Revilla, Enette Mae, et al. "Adaptive data-driven motion detection and optimized correction for brain PET." Neuroimage 252 (2022): 119031.

Comment 3: Figures are hard to read and there are no captions. Consider adding the captions for the figures, and make the figure self-explanatory. Replace Figure 5 and 6 with vector graphics.

Comment 4: There is no comparison between Vicra motion tracking result and data-driven motion tracking. The author may consider including the quantitative measurements between the distance of the two measurements.

Comment 5: Page 10, line 235. Reference 17 and 18 are old references, the author should consider reporting the latest research for the markerless motion tracking.

Slipsager, Jakob M., et al. "Markerless motion tracking and correction for PET, MRI, and simultaneous PET/MRI." PloS one 14.4 (2019): e0215524.

Comment 6: In the discussion, the author may want to discuss the recent deep learning data-driven methods for brain PET head motion correction. For example,

[1] Tumpa, Tasmia Rahman, et al. "Deep learning based registration for head motion correction in positron emission tomography as a strategy for improved image quantification." Frontiers in Physics 11 (2023): 246.

[2] Zeng, Tianyi, et al. "Fast Reconstruction for Deep Learning PET Head Motion Correction." International Conference on Medical Image Computing and Computer-Assisted Intervention. Cham: Springer Nature Switzerland, 2023.

[3] Zeng, Tianyi, et al. "Supervised Deep Learning for Head Motion Correction in PET." International Conference on Medical Image Computing and Computer-Assisted Intervention. Cham: Springer Nature Switzerland, 2022.

6. PLOS authors have the option to publish the peer review history of their article (what does this mean?). If published, this will include your full peer review and any attached files.

Reviewer #1: No

Reviewer #2: No

---

## [Author Response · Author response to Decision Letter 0]

11 Mar 2024

Reviewers' Comments to the Authors:

Dear Reviewer ＃１

Thank you very much for reviewing our manuscript and offering valuable advice. We have addressed your comments with point-by-point responses and revised the manuscript accordingly.

１. Abstract. A big limitation of the study is that the method was only validated with Hoffman phantom but not human brain. This should be stated explicitly.

Author response: We have made some modifications to the introduction and conclusion parts of the abstract as follows.

Introduction: In this study, we investigated the effectiveness of data-driven brain motion correction techniques on positron emission tomography/computed tomography images using a Hoffman phantom, involving continuous rotational motion and tilting motion, each expanded up to approximately 20 degrees. ( page 2, lines 41-43)

Conclusions: However, a significant limitation of this study is the exclusive validation of the proposed method using a Hoffman phantom; its applicability to the human brain has not been investigated. Further research involving human subjects is necessary to assess the generalizability and reliability of the presented motion correction technique in real clinical scenarios. (Page 3, lines 63-67)

２. Abstract. “% contrast and normalized mean squared error were improved after correction.” Numbers would talk more than simply stating “improved”.

Author response: We have included NMSE values and additional analyzed SSIM values in the results section of the abstract.

Normalized Mean Squared Error (NMSE) results demonstrated the effectiveness of DDBMC in compensating for rotational and tilting motions during PET imaging. In Cases 1 and 2 involving rotational motion, NMSE decreased from 0.15-0.2 to approximately 0.01 with DDBMC, indicating a substantial reduction in differences from the reference image across various brain regions. In the Structural Similarity Index (SSIM), DDBMC improved it to above 0.96 Contrast assessment revealed notable improvements with DDBMC. In continuous rotational motion, % contrast increased from 42.4% to 73.5%, In tilting motion, % contrast increased from 52.3% to 64.5%, eliminating significant differences from the static reference image. These findings underscore the efficacy of DDBMC in enhancing image contrast and minimizing motion induced variations across different motion scenarios. (Page 2, lines 50 - Page 3, lines 60)

３. Abstract. “Although the effectiveness of brain motion correction was confirmed in this experiment, it is necessary to understand the relationship between the range of motion and limitations in brain motion correction processing because there might be more complex movements in clinical practice.” I consider you are not saying results. Please move to conclusions.

Author response: We have revised the concluding sentence in light of the significant content of your proposal. 

DDBMC processing can effectively compensate for continuous rotational and tilting motion of the head during PET, with motion angles of approximately 20 degrees. However, a significant limitation of this study is the exclusive validation of the proposed method using a Hoffman phantom; its applicability to the human brain has not been investigated. Further research involving human subjects is necessary to assess the generalizability and reliability of the presented motion correction technique in real clinical scenarios. (Page 3, lines 61-67)

４. Introduction. “tau PET imaging preparations are also being investigated.” What is “imaging preparations”? Please define.

Author response: Tau PET tracers such as F-18-PM-PBB3 and F-18-SNFT-1 are being studied domestically and internationally, with anticipated clinical applications in the future [6, 7]. (Page 3, lines 76-78)

［6］Endo H, Tagai K, Ono M, Ikoma Y, Oyama A, Matsuoka K, et al. A Machine Learning-Based Approach to Discrimination of Tauopathies Using [(18) F]PM-PBB3 PET Images. Mov Disord. 2022;37(11):2236-46. Epub 20220828. doi: 10.1002/mds.29173. PubMed PMID: 36054492; PubMed Central PMCID: PMCPMC9805085.

［7］Harada R, Lerdsirisuk P, Shimizu Y, Yokoyama Y, Du Y, Kudo K, et al. Preclinical Characterization of the Tau PET Tracer [(18)F]SNFT-1: Comparison of Tau PET Tracers. J Nucl Med. 2023;64(9):1495-501. Epub 20230615. doi: 10.2967/jnumed.123.265593. PubMed PMID: 37321821.

５. Introduction. “Movement during the examination might result in inaccurate images.” An image cannot be inaccurate, only its quantification can.

Author response: We have added and revised the citation references.

The most cited effects of patient motion are frame image misalignment, which affects the dynamic analysis of dynamic protocols, and resolution loss due to motion blurring in the frame. Moreover, motion during scans can cause mis-estimation in tracer kinetic modeling, introducing inaccuracies in the quantitative analysis of the data [15]. (Page 4, lines 85-89)

［15］Anton-Rodriguez JM, Sibomana M, Walker MD, Huisman MC, Matthews JC, Feldmann M, et al., editors. Investigation of motion induced errors in scatter correction for the HRRT brain scanner2010: IEEE.

６. Introduction. “Research and development of technologies to compensate for head motion is underway, including hardware-based motion tracking methods that track markers attached to the head with an external device…” Hardware based motion measure and compensation methods have been mature for more than a decade. So it is not appropriate to say they are still underway.

Author response: We have added and revised the citation references.

Several techniques have been implemented to compensate for head motion, including hardware-based motion tracking methods that utilize an external device to track a marker attached to the head [15, 16], and marker-less tracking methods that employ a charged-coupled device camera to extract the surface shape of the object [17, 18]. (Page 4, lines 89-93)

［17］Zeng T, Lu Y, Jiang W, Zheng J, Zhang J, Gravel P, et al. Markerless head motion tracking and event-by-event correction in brain PET. Phys Med Biol. 2023;68(24). Epub 20231212. doi: 10.1088/1361-6560/ad0e37. PubMed PMID: 37983915; PubMed Central PMCID: PMCPMC10713921.

［18］Iwao Y, Akamatsu G, Tashima H, Takahashi M, Yamaya T. Brain PET motion correction using 3D face-shape model: the first clinical study. Ann Nucl Med. 2022;36(10):904-12. Epub 20220719. doi: 10.1007/s12149-022-01774-0. PubMed PMID: 35854178; PubMed Central PMCID: PMCPMC9515015

７. Introduction. “charged-coupled device(CCD)” No need to state the abbreviation if it is only mentioned once.

Author response: The abbreviation has been removed.

charged-coupled device camera (Page 4, lines 92)

８. Introduction. “A data-driven brain motion correction (DDBMC) technique…” Although reader can find implementation details themselves, it is still helpful to introduce how it works briefly.

Author response: We have added and revised the citation references.

In DDBMC, image reconstruction is performed with ultra-short frames (≤1 second) from listmode data . Motion is estimated by performing image rigid registration between each frame and a selected reference frame, allowing for motion-corrected listmode reconstruction. Each frame is reconstructed using maximum-likelihood expectation maximization without subsets, because of the low number of events in each frame. Since accurate quantification is not crucial, scatter correction and point spread function modeling are omitted, and only a few iterations are carried out. If necessary, you can use the standard attenuation correction map generated from CT images[5, 23]. (Page 4, lines97- Page 5. lines104)

［23］Revilla EM, Gallezot JD, Naganawa M, Toyonaga T, Fontaine K, Mulnix T, et al. Adaptive data-driven motion detection and optimized correction for brain PET. Neuroimage. 2022;252:119031. Epub 20220304. doi: 10.1016/j.neuroimage.2022.119031. PubMed PMID: 35257856; PubMed Central PMCID: PMCPMC9206767.

９. Introduction. “In the present study, we validated our method…” Consider change “our” to “the” as it is not proposed by the authors.

Author response: we validated the method using PET/CT (Page 5, lines 105)

１０. Materials and methods. GE HealthCare - GE Healthcare

Author response: Since the division in January 2023, The company name has been changed to GE HealthCare. [Link: https://www.gehealthcare.com/]

We will use the notation 'GE HealthCare' this time. (Page 5, lines 113)

１１. Lutetium based scintillators (LBSs). Do you mean LSO or something else?

Author response: The following correction is made.　

It consists of a 128-slice CT system and a 4-ring PET system with lutetium-yttrium oxyorthosilicate (LYSO) crystals and silicon photomultiplier-based (SiPM) detectors providing a 20-cm axial field of view and a 70-cm transaxial field of view. (Page 5, lines 114-117)

１２. “The LBSs and SiPMs detectors enable time resolution below 380 ps and time-of-flight compatibility.” Consider rephrasing it to “The LBSs and SiPMs detectors enable time-of-flight compatibility with time resolution below 380 ps.” I did not go through the whole paper, but it is likely other sentences need to be rephrased too.

Author response: We will correct as you indicated.

The LYSO and SiPM detectors enable time-of-flight compatibility with time resolution below 380 ps. [24, 25]. (Page 55, lines 119-120)

１３. “Although the range of motion is considered sufficient, it is assumed that a variety of movements will be mixed in actual.” Although you simulated three cases of movement, they still cannot represent the real case. For example, a real head movement can be the combination of six DOFs. Can you comment on this?

Author response: We have added this as a limitation to this study.

In clinical examinations, head motion involves a combination of six degrees of freedom (6-DOF). Spangler-Bickell et al. reported the effectiveness of DDBMC even in clinical cases [20]. This suggests that DDBMC is useful for various movements. However, understanding the limits of the correction region for movements based on the combination of 6-DOF is crucial, and further verification is deemed necessary in the future. (Page 18, lines 397-402)

１４. Line 133, 1 second/frame was used. Have you performed denoising? “list data” should be “listmode data”.

Author response: The reconstruction method for 1-second data does not include noise removal. 

For reconstruction of the ultra-short frames, the maximum likelihood expectation maximization method was used. Reconstructions were performed with six iterations, with no attenuation or scatter corrections [5]. (Page 7, lines 172- Page 8, lines 174)

We have made changes to all instances where “list data”was mentioned and replaced it with “listmode data.” (page 4, line 100, page 7, line 170, page 10, line 225,227, 234, 238, page 18, line 387)

１５. Contrast and NMSE are good metrics to evaluate the correction. It would be nice to see SSIM to reflect the recovery of global features.

Author response: As you suggested, we have incorporated SSIM analysis as an image evaluation method and made the following modifications.

Method: Furthermore, a structure similarity index (SSIM) was used to measure the degree of similarity between the reference image and an image of interest [28]. The SSIM is sensitive to contrast, luminance, and structures within an image, with values ranging from 0 to 1. As SSIM approaches 1, the similarity with the reference image increases. Of the total of Seventy-one slices from one-bed scan imaging, it evaluated 43 slices delineated from the parietal lobe to the cerebellum. SSIM calculations were performed using skimage 0.21.0 and Python 3.8.10. (Page 8, lines 194- Page 9, lines 200)

Result: Normalized mean squared error and structure similarity index

The results of NMSE for each slice, comparing images with rotational or tilting motion to the reference images taken in a static state, are presented in Fig 7. Additionally, Table 2. provides the mean and standard deviation values of NMSE and SSIM for all slices. In the case of images with rotational motion (Cases 1 and 2), NMSE ranged from 0.15 to 0.2 from the basal ganglia to the cerebellum, indicating a significant difference from the reference images. With DDBMC, the variation based on slice position and reduces differences from the reference images. With DDBMC, mean NMSE decreases to approximately 0.01 for both Cases 1 and 2. In terms of SSIM, there is a significant improvement, with Case 1 increasing from 0.892 to 0.968, and Case 2 increasing from 0.860 to 0.967. In Case 3, involving tilting motion, the differences in NMSE values in the basal ganglia were smaller than those in rotational motion. However, the NMSE values in the vertex and cerebellar regions were larger. With DDBMC, NMSE significantly improved from 0.262 to 0.016, and SSIM increased from 0.874 to 0.962. (Page 13, lines 289- 301)

Discussion: Furthermore, the SSIM with DDBMC showed a significantly higher value of 0.96 or higher than that without DDBMC. We believe that this result is consistent with the fact that all cases after DDBMC as "Diagnostic" in the visual evaluation by the specialist physicians. (Page 17, lines 371-374)

１６. “The results of the visual evaluation are shown in Table 1” Four average scores are 4, which suggests all raters have consensus of scoring 4? Without is out of boundary.

Author response: The text has been revised due to unclear content, and furthermore, Table 1 has been added.

In images without DDBMC, the average scores were 0.2 for Case 1 and 1.0 for Case 2, indicating “Not diagnostic”. For Case 3, the average score was 2.6. All evaluators judged the reference images and images with DDBMC as “Diagnostic”. (Page 12, lines 278-281)

１７. “The closer NMSE is to 0, the closer the image with motion is to the reference image” Consider remove this sentence, as it already appeared in Methods.

Author response: We have removed the part you indicated.

１８. Discussion. “However, it is difficult for the subject to remain still because of the long time needed to acquire images”. New scanners can scan head within a minute. The associated challenge will then be the vulnerability of ultra-high resolution.

Author response: We have made the following corrections.

Because head movements during brain PET imaging can lead to image degradation, the head is immobilized during imaging. However, due to the extended time required for image acquisition, it can be challenging for all subjects to remain completely still, depending on their age and physical condition. In short-duration imaging, there is a potential for decreased resolution. Therefore, techniques are needed to minimize the effect of head motion on the images. (Page 15, lines 327- Page 16, 332)

１９. “In this study, we used DDBMC, which was devised as a correction method that does not require an external device.” Some important references are missing. There are other data-driven method papers in recent years. For example:Rezaei et al. PMB 2021, Rigid motion tracking using moments of inertia in TOF-PET brain studies. Sun T. et al. PMB 2022, An iterative image-based inter-frame motion compensation method for dynamic brain PET imaging.

Author response: As suggested, we have added the article to the Reference.

A data-driven brain motion correction (DDBMC) technique has been developed [19-22]. (Page 4, lines 96-97)

[21]Rezaei A, Spangler-Bickell M, Schramm G, Van Laere K, Nuyts J, Defrise M. Rigid motion tracking using moments of inertia in TOF-PET brain studies. Phys Med Biol. 2021;66(18). Epub 20210913. doi: 10.1088/1361-6560/ac2268. PubMed PMID: 34464941.

[22]Sun T, Wu Y, Bai Y, Wang Z, Shen C, Wang W, et al. An iterative image-based inter-frame motion compensation method for dynamic brain PET imaging. Phys Med Biol. 2022;67(3). Epub 20220202. doi: 10.1088/1361-6560/ac4a8f. PubMed PMID: 35021156.

２０. A limitation is the study was only validated in static scan but not dynamic scan.

Author response: Added as a limitation to this study.

The results of this study mainly apply to static data, where changes in tracer distribution over time are minimal. In dynamic data, which captures changes in tracer distribution, it is necessary to update the reference image regularly. The optimal strategy for handling thi

---

## [Decision Letter · Decision Letter 1]

17 Mar 2024

PONE-D-24-01135R1Verification of the effect of Data-Driven Brain Motion Correction on PET imagingPLOS ONE

Dear Dr. Odagiri,

Thank you for submitting your manuscript to PLOS ONE. After careful consideration, we feel that it has merit but does not fully meet PLOS ONE’s publication criteria as it currently stands. Therefore, we invite you to submit a revised version of the manuscript that addresses the points raised during the review process.

We look forward to receiving your revised manuscript.

Kind regards,

Khan Bahadar Khan, Ph.D

Academic Editor

PLOS ONE

Journal Requirements:

Reviewers' comments:

Reviewer's Responses to Questions

**Comments to the Author**

1. If the authors have adequately addressed your comments raised in a previous round of review and you feel that this manuscript is now acceptable for publication, you may indicate that here to bypass the “Comments to the Author” section, enter your conflict of interest statement in the “Confidential to Editor” section, and submit your "Accept" recommendation.

Reviewer #1: (No Response)

Reviewer #2: (No Response)

2. Is the manuscript technically sound, and do the data support the conclusions?

Reviewer #1: Partly

Reviewer #2: Yes

3. Has the statistical analysis been performed appropriately and rigorously? 

Reviewer #1: Yes

Reviewer #2: Yes

4. Have the authors made all data underlying the findings in their manuscript fully available?

Reviewer #1: Yes

Reviewer #2: Yes

5. Is the manuscript presented in an intelligible fashion and written in standard English?

Reviewer #1: Yes

Reviewer #2: Yes

6. Review Comments to the Author

Reviewer #1: Thank you for addressing most of my comments. My only remaining concern is for comment no. 14. Do you have an example image? And for a patient scan, the image noise for 1-sec frame could be larger than you ideal phantom image.

Reviewer #2: 1. Revised manuscript line 94: The hardware-based motion tracking requires a tracking device and additional setup, but it does not require a device to be attached to the patient. Usually, the device was fixed on the PET scanner gantry. The author should correct this sentence.

2. Revised manuscript line 120: references 24 and 25 are related to the reconstruction methodology. The author should consider removing them and adding the system performance paper of the Discovery PET for reference.

3. Revised manuscript line 97: reference 22 is not related to the DDBMC. The author should consider correcting this error. In addition, the other data-driven MC methods should be introduced, like COD (reference 23) and method in reference 22.

4. Revised manuscript line 417: reference 17 is a device-based motion correction method. Here the author wants to discuss deep learning method for PET head motion correction, I would recommend reference 31. The other possible reference related to the topic:

Reimers, Erik, Ju-Chieh Kevin Cheng, and Vesna Sossi. "Deep Learning Aided Intra-Frame Motion Correction for Low-Count Dynamic Brain PET." IEEE Transactions on Radiation and Plasma Medical Sciences (2023).

Zeng, Tianyi, et al. "Supervised deep learning for head motion correction in PET." International Conference on Medical Image Computing and Computer-Assisted Intervention. Cham: Springer Nature Switzerland, 2022.

7. PLOS authors have the option to publish the peer review history of their article (what does this mean?). If published, this will include your full peer review and any attached files.

Reviewer #1: No

Reviewer #2: No

---

## [Author Response · Author response to Decision Letter 1]

20 Mar 2024

Reviewers' Comments to the Authors:

Dear Reviewer ＃１

Thank you very much for reviewing our manuscript and offering valuable advice. 　We have addressed your comments with point-by-point responses and revised the manuscript accordingly.

1. Thank you for addressing most of my comments. My only remaining concern is for comment no. 14. Do you have an example image? And for a patient scan, the image noise for 1-sec frame could be larger than you ideal phantom image.

Author response: Thank you for your suggestion. We have added a reference [20] showing 1-second images in clinical data.

DDBMC makes ultra-short (approximately 1 second/frame, depending on the data) reconstructions from listmode data and estimates motion between these ultra-short images and a reference image at the beginning of acquisition (30 seconds/frame) using rigid registration [20]. ( page 7, lines 173)

Dear Reviewer ＃2

Thank you very much for reviewing our manuscript and offering valuable advice. 　We have addressed your comments with point-by-point responses and revised the manuscript accordingly.

1. The hardware-based motion tracking requires a tracking device and additional setup, but it does not require a device to be attached to the patient. Usually, the device was fixed on the PET scanner gantry. The author should correct this sentence.

Author response: Due to the confusing wording, we have revised it as follows.

The motion tracking method using Polaris Vicra requires the attachment of light-reflective markers to the patient, which is time-consuming to set up and may cause the attached markers to come off due to movement [19]. ( page 4, lines 93-96)

[19]. Woo S-K, Watabe H, Choi Y, Kim KM, Park CC, Bloomfield PM, et al. Sinogram-based motion correction of pet images using optical motion tracking system and list-mode data acquisition. IEEE Transactions on Nuclear Science. 2004;51(3):782-8.

2. references 24 and 25 are related to the reconstruction methodology. The author should consider removing them and adding the system performance paper of the Discovery PET for reference.

Author response: We have added references "24, and 25" regarding the performance of the PET scanner.　 ( page 5, lines 121)

[24].Hsu DFC, Ilan E, Peterson WT, Uribe J, Lubberink M, Levin CS. Studies of a Next-Generation Silicon-Photomultiplier-Based Time-of-Flight PET/CT System. J Nucl Med. 2017;58(9):1511-8. Epub 20170427. doi: 10.2967/jnumed.117.189514. PubMed PMID: 28450566.

[25]. Chicheportiche A, Marciano R, Orevi M. Comparison of NEMA characterizations for Discovery MI and Discovery MI-DR TOF PET/CT systems at different sites and with other commercial PET/CT systems. EJNMMI Phys. 2020;7(1):4. Epub 20200114. doi: 10.1186/s40658-020-0271-x. PubMed PMID: 31938953; PubMed Central PMCID: PMCPMC6960280.

3. reference 22 is not related to the DDBMC. The author should consider correcting this error. In addition, the other data-driven MC methods should be introduced, like COD (reference 23) and method in reference 22.

Author response: Reference 22 has been removed. ( page 4, lines 98)

The following sentence was added to the discussion.

Revilla et al. proposed a data-driven method, the three-dimensional center of distribution, reporting significant reduction in blurring due to head motion [23]. It has been noted that this method may not be able to accommodate large changes in tracer activity within the brain. Verification is needed in the present study regarding the DDBMC examined here in cases of significant changes in tracer activity within the brain. ( page 19, lines 414-419)

4. reference 17 is a device-based motion correction method. Here the author wants to discuss deep learning method for PET head motion correction, I would recommend reference 31. The other possible reference related to the topic:

Reimers, Erik, Ju-Chieh Kevin Cheng, and Vesna Sossi. "Deep Learning Aided Intra-Frame Motion Correction for Low-Count Dynamic Brain PET." IEEE Transactions on Radiation and Plasma Medical Sciences (2023).

Zeng, Tianyi, et al. "Supervised deep learning for head motion correction in PET." International Conference on Medical Image Computing and Computer-Assisted Intervention. Cham: Springer Nature Switzerland, 2022.

Author response: Reference 17 has been removed, and we have added reference "34" as a relevant citation. ( page 19, lines 423)

[34]. Zeng T, Zhang J, Revilla E, Lieffrig EV, Fang X, Lu Y, et al. Supervised Deep Learning for Head Motion Correction in PET. Med Image Comput Comput Assist Interv. 2022;13434:194-203. Epub 20220916. doi: 10.1007/978-3-031-16440-8_19. PubMed PMID: 38107622; PubMed Central PMCID: PMCPMC10725740.

We would like to express our sincere gratitude for your invaluable time, expertise, and thoughtful feedback during the review process. We look forward to your continued support in the future.

---

## [Decision Letter · Decision Letter 2]

26 Mar 2024

Verification of the effect of Data-Driven Brain Motion Correction on PET imaging

PONE-D-24-01135R2

Dear Dr. Odagiri,

We’re pleased to inform you that your manuscript has been judged scientifically suitable for publication and will be formally accepted for publication once it meets all outstanding technical requirements.

Kind regards,

Khan Bahadar Khan, Ph.D

Academic Editor

PLOS ONE

Additional Editor Comments (optional):

Reviewers' comments:

Reviewer's Responses to Questions

**Comments to the Author**

1. If the authors have adequately addressed your comments raised in a previous round of review and you feel that this manuscript is now acceptable for publication, you may indicate that here to bypass the “Comments to the Author” section, enter your conflict of interest statement in the “Confidential to Editor” section, and submit your "Accept" recommendation.

Reviewer #1: All comments have been addressed

Reviewer #2: All comments have been addressed

2. Is the manuscript technically sound, and do the data support the conclusions?

Reviewer #1: Yes

Reviewer #2: Yes

3. Has the statistical analysis been performed appropriately and rigorously? 

Reviewer #1: Yes

Reviewer #2: Yes

4. Have the authors made all data underlying the findings in their manuscript fully available?

Reviewer #1: Yes

Reviewer #2: Yes

5. Is the manuscript presented in an intelligible fashion and written in standard English?

Reviewer #1: Yes

Reviewer #2: Yes

6. Review Comments to the Author

Reviewer #1: (No Response)

Reviewer #2: (No Response)

7. PLOS authors have the option to publish the peer review history of their article (what does this mean?). If published, this will include your full peer review and any attached files.

Reviewer #1: No

Reviewer #2: No

---

## [Editor Report · Acceptance letter]

15 May 2024

PONE-D-24-01135R2 

PLOS ONE

Dear Dr. Odagiri, 

I'm pleased to inform you that your manuscript has been deemed suitable for publication in PLOS ONE. Congratulations! Your manuscript is now being handed over to our production team.

Kind regards, 

on behalf of

Dr. Khan Bahadar Khan 

Academic Editor

PLOS ONE